# Risk factors of child mortality in Ethiopia: Application of multilevel two-part model

Setegn Muche Fenta[1]*, Haile Mekonnen Fenta[2]

**1** Department of Statistics, Faculty of Natural and Computational Sciences, Debre Tabor University, Debre Tabor, Ethiopia, **2** Department of Statistics, College of Science, Bahir Dar University, Bahir Dar, Ethiopia

\* setegn14@gmail.com

**Data Availability Statement:** We have used women data set of EDHS 2016 for this study. The data set was accessed from the Measure DHS website (http://www.measuredhs.com).

## Abstract

The child mortality rate is an essential measurement of socioeconomic growth and the quality of life in Ethiopia which is one among the six countries that account for half of the global under-five deaths. Therefore, this study aimed to identify the potential risk factors for child mortality in Ethiopia. Data for the study was drawn from the Ethiopian Demographic and Health Survey data conducted in 2016. A two-part random effects regression model was employed to identify the associated predictors of child mortality. The study found that 53.3% of mothers did not face any child death, while 46.7% lost at least one. Vaccinated child (IRR = 0.735, 95%CI: 0.647, 0.834), were currently using contraceptive (IRR = 0.885, 95% CI: 0.814, 0.962), who had antenatal care visit four or more times visit (IRR = 0.841, 95%CI: 0.737,0.960), fathers whose level of education is secondary or above(IRR = 0.695, 95%CI: 0.594, 0.814), mothers who completed their primary school(IRR = 0.785, 95%CI: 0.713, 0.864), mothers who have birth interval greater than 36 months (IRR = 0.728, 95%CI: 0.676, 0.783), where the age of the mother at first birth is greater than 16 years(IRR = 0.711, 95% CI: 0.674, 0.750) associated with the small number of child death. While multiple births (IRR = 1.355, 95%CI: 1.249, 1.471, four and above birth order (IRR = 1.487, 95%CI: 1.373, 1.612) and had working father (IRR = 1.125, 95%CI: 1.049, 1.206) associated with a higher number of child death. The variance components for the random effects showed significant variation of child mortality between enumeration areas. Policies and programs aimed at addressing enumeration area variations in child mortality need to be formulated and their implementation must be strongly pursued. Efforts are also needed to extend educational programmers aimed at educating mothers on the benefits of the antenatal checkup before first birth, spacing their birth interval, having their child vaccinated, and selecting a safe place of delivery to reduce child mortality.

## Introduction

The child mortality rate is an indicator of child health as well as the overall development and well-being of a population. As part of their Sustainable Development Goals (SDG), the United Nations has set a target of reducing child mortality to as low as 25 per 1000 live births by 2030

**Funding:** The authors received no specific funding for this work.

**Competing interests:** The authors have declared that no competing interests exist.

[1–3]. According to different scholars and organizations, child mortality conditions remain the same, over 60 million children will die until 2030 [2–4]. The objective of the Millennium Development Goals (MDG) 4 was to reduce mortality in children under five years by two-thirds. Nevertheless, achieving this goal is hampered by the limited availability of data about accurate estimates of their mortality rate [5, 6]. Of the 5.3 million deaths of the under-five children in 2018 more than 50% (3.3 million) were in Sub-Saharan Africa. Fifty percent of the deaths occurred in six countries including Ethiopia and 15000 children die every single day globally. Most children died of preventable or treatable causes such as complications during birth, pneumonia, diarrhea, neonatal sepsis, and malaria [1–4, 7].

Ethiopia has one of the highest rates of child deaths and disabilities in the world. More than 704 children die every day from easily preventable diseases [3, 8]. If situations continue as such, more than 3,084,000 children will die until 2030. In Ethiopia, there have been regional variations in child mortality [3, 9, 10]. Child mortality rates range from as low as 39 per 1,000 live births in Addis Ababa to as high as 125 per 1,000 live births in Afar [9].

The government of Ethiopia is struggling to minimize the death of under-five children, henceforth (U5C), and there had been improvements over the past years. Despite the progress, geographical locations, health services, maternal socioeconomic characteristics, etc, still pose challenges. Hence, identification of enumeration area-specific determinants on the number of U5C mortality per mother is crucial to plan and implement interventions and take actions to address the burden of mortality of U5C in Ethiopia.

Previous studies in developing countries on child mortality considered either prevalence alone (i.e. whether the death occurred in a household) and used logistic and survival models to analyze such [11–14], or severity only (i.e. compared the number of reported deaths) and applied count regression models [10]. In this study, however, we proposed that there are two processes: whether the death occurred in the household (prevalence part) and the number of reported deaths if death did occur (severity part). The two-part model has extensive (the zeros) and intensive (the positives) margins in a multi-index count model, representing nonoccurrence and occurrence of death, respectively. It is also referred to as a zero-hurdle model because it allows for a systematic difference in the statistical process governing individuals (observations) below the hurdle and individuals above the hurdle set at zero. An alternative approach to the two-part process is finite mixtures, a combination of zeros point mass distribution and the nonzero distribution [15, 16]. Most works in this area; however, do not address concerns that occurrence of death (prevalence) and the number of deaths (severity) reported in a household are joint processes, and that failing to account for the joint nature of these processes and complex sampling method [17] can bias estimates of risk factors on child mortality [10].

Furthermore, the variation in the determinants of the number of child mortality may be due to heterogeneity in the enumeration area of the study. To address this, we proposed a two-part random effects model for child mortality data [18, 19]. The proposed model consisted of two generalized linear mixed models (GLMM) with correlated random effects; the first part assumed a GLMM with a logistic link and the second part explored a count model negative binomial distribution. Therefore, this study aimed to identify the factors associated with the number of child mortality in Ethiopia.

## Materials and methods

### Data source and study design

The data for this study was obtained from the 2016 Ethiopian Demographic Health Survey (EDHS), particularly data on birth record data which is a population-based, cross-sectional

survey of a complex sampling design involving region and residence as strata. The first stage of the selection was 645 PSU with 202 EAs urban and 443 EAs rural areas based on the 2007 Ethiopian Population and Housing Census (PHC) of the Ethiopian Central Statistics Agency (CSA). From a total of 18,008 households 16,650 having a response rate of 98% of the response rate households were eligible. The women were interviewed for information on their birth history questioners and a total of 14370 births were considered for this study (the EDHS 2016 can be accessed on request through proper format).

## Ethics statement

This study is a secondary data analysis of the EDHS, which is publicly available, approval was sought from MEASURE DHS/ICF International, and permission was granted for this use. The original DHS data were collected in conformity with international and national ethical guidelines. Ethical clearance was provided by the Ethiopian Public Health Institute (EPHI) (formerly the Ethiopian Health and Nutrition Research Institute (EHNRI) Review Board, the National Research Ethics Review Committee (NRERC) at the Ministry of Science and Technology, the Institutional Review Board of ICF International, and the United States Centers for Disease Control and Prevention (CDC). Written consent was obtained from mothers/caregivers and data were recorded anonymously at the time of data collection during the EDHS 2016.

## Variable of the study

**Dependent variable.** The dependent variable was the number of child deaths per mother, a counted outcome from birth record dataset in the EDHS 2016. Region, mother's age, the education level of the father, education level of the mother, father's occupation, mother's occupation, family size, age of mother at first birth, religion, vaccination of child, contraceptive use, birth order, preceding birth interval, child twin, place of delivery, antenatal visit, breastfeeding, and residence were the potential predictors of child death.

The secondary data were managed in SPSS software version 21and then exported to R version 3.5.3 for analysis

## Statistical analysis

Child death was better envisaged as a two-part model of two types:—Zero-Inflated and the Zero-Hurdle models. Zero-inflated models allow overdispersion as well as zero-inflated count data. The frequently used models for zero-inflated count data are zero-inflated Poisson (ZIP) and zero-inflated negative binomial (ZINB). The ZIP model, introduced by Lambert, D. et al. [20], provides a dual-state method for modeling data characterized by a significant amount or more zeros than would be expected in a traditional Poisson or negative binomial model, while the ZINB model, introduced by Greene, W.H. et al. [21], is a more flexible one that handles over-dispersion caused by both unobserved heterogeneity and excess zeroes. Zero-inflated regression considers two data generating processes and assumes zero counts coming from two different sources from the always-zero group (mothers who are never born) or the not-always-zero group (mothers who may not be dead her child). Zero-inflated regression is a two-part model. A Logit model determines if a zero count is from the always-zero group or the not-always-zero group, and a baseline model, whether Poisson or Negative binomial, governs both zero and positive counts from the not-always-zero group [22]. ZIP regression is useful for modeling count data with excess zeros, however, in hierarchical study design and data collection procedure, zero-inflation and correlation may occur simultaneously [23]. Multilevel ZIP

regression has been employed to overcome these problems. For the ZIP models, we have

$$p(Y_{ij} = y_j) = \begin{cases} \pi_{ij} + (1 - \pi_{ij})\exp(-\mu), & if \ y_{ij} = 0 \\ (1 - \pi_{ij})\dfrac{\exp(-\mu)\mu^{y_{ij}}}{y_{ij}!}, & if \ y_{ij} = 1, 2, \ldots\ldots \end{cases} \qquad 0 \le \pi_{ij} \le 1 \ (1)$$

Where $Y_{ij}$ indicates the number of under-five death the $i^{th}$ mother in the $j^{th}$ enumeration area and $\mu$ is the mean for the Poisson distribution. If over-dispersion is attributed to factors beyond the inflation of zeros, a ZINB model is more appropriate [24]. A multilevel ZINB regression incorporating random effects to account for data dependency and over-dispersion is used [25]. Let $Y_{ij}(i = 1,2,\ldots\ldots,n; j = 1,2,\ldots\ldots,m)$ be a count say, the under-five death of the $i^{th}$ mother in $j^{th}$ the enumeration area follows a ZINB distribution:

$$p(Y_{ij} = y_{ij}) = \begin{cases} \pi_{ij} + \dfrac{(1 - \pi_{ij})}{(1 + \alpha\mu)^{-\frac{1}{\alpha}}}, & if \ y_{ij} = 0 \\ 1 - \pi_{ij}\dfrac{\Gamma(y_{ij} + {}^{1}\!/_{\alpha})}{y_{ij}!\Gamma({}^{1}\!/_{\alpha})}(1 + \alpha\mu_{ij})^{-\frac{1}{\alpha}}\left(1 + \dfrac{1}{\alpha\mu}\right)^{-y_{ij}}, & if \ y_{ij} > 0 \end{cases} \qquad 0 \le \pi_{ij} \le 1 \ (2)$$

With parameters $\mu \ge 0$ for the mean and $\alpha > 0$ over-dispersion
Then the two-level ZINB and ZIP regression model can be expressed in vector form as:

$$\log(\mu_{ij}) = \beta_o + \sum_{l=1}^{k} \beta_l x_{lij} + U_{oj} + \sum_{l=1}^{k} U_{lj} x_{lij} \qquad (3)$$

$$\log it(\pi_{ij}) = \log\left(\frac{\pi_{ij}}{1 - \pi_{ij}}\right) = \gamma_o + \sum_{l=1}^{k} \gamma_l z_{lij} + W_{oj} + \sum_{l=1}^{k} W_{lj} z_{lij} \qquad (4)$$

Here, the covariates $X_{ij}$ and $Z_{ij}$ appearing in the respective negative binomial and logistic components are not necessarily the same $\beta$ and $\gamma$ are the corresponding vectors of regression coefficients [25, 26]. The vectors $w_j$ $u_j$ and denote the enumeration area-specific random effects for simplicity of presentation. The random effect $u$ and $w$ are assumed to be independent and normally distributed with a mean of zero and variance of $\sigma_u^2$ and $\sigma_w^2$ respectively. A special case of the above models is the zero-hurdle model. The hurdle regression handles the excess zeros by relaxing the assumption that zeros and positives come from a single data generating process [15]. The hurdle model is flexible in handling both under and overdispersion problems. A hurdle model is introduced by [16] for the analysis of over-dispersed or under-dispersed count data. The hurdle model, like the ZI model approach, is a 2-part count regression method that deals with excess zeros in the data. However, hurdle models are different from ZI in that its first component contains: -a binomial distribution that determines if a count is zero or positive. The second part is truncated at zero models governing the positive counts, i.e. $E(Y_i/Y_i > 0)$ [15]. Poisson Hurdle model can be written as follows

$$p(Y_{ij} = y_{ij}) = \begin{cases} \pi_{ij} & if \ y_{ij} = 0 \\ (1 - \pi_{ij})\dfrac{\exp(-\mu)\mu^{y_{ij}}}{(1 - \exp(-\mu))y_{ij}!} & if \ y_{ij} = 1, 2, \ldots \quad .. \end{cases} \qquad 0 \le \pi_{ij} \le 1 \ (5)$$

Where $Y_{ij}$ is the number of under-five death for the $i^{th}$ mother in the $j^{th}$ enumeration area $\mu_{ij}$ is the mean and $\pi_{ij}$ is zero proportion parameters. An alternative to the Poisson hurdle is a

negative binomial given by the Eq (5)

$$
p(Y_{ij} = y_{ij}) = \begin{cases} \pi_{ij}, & \text{if } y_{ij} = 0 \\ (1 - \pi_{ij}) \dfrac{\Gamma(y_{ij} + {}^{1}/_{\alpha})}{y_{ij}!\Gamma({}^{1}/_{\alpha})}(1 + \alpha\mu_{ij})^{-\frac{1}{\alpha}}\left(1 + \dfrac{1}{\alpha\mu}\right)^{-y_{ij}}, & \text{if } y_{ij} > 0 \end{cases} \quad 0 \le \pi_{ij} \le 1 \tag{6}
$$

with parameters $\mu \geq 0$ for the mean and $\alpha > 0$ for over-dispersion

In the regression setting, both the mean $\mu_{ij}$ and zero proportion $\pi_{ij}$ parameters are related to the covariate vectors $x_{ij}$ and $z_{ij}$ respectively. Moreover, responses within the same enumeration area are likely to be correlated. To accommodate the inherent correlation, random effects $u_j$ and $w_j$ are incorporated in the linear predictors $\eta_{ij}$ for the Poisson part and $\xi_{ij}$ for the zero part. The Poisson Hurdle and Negative binomial Poisson mixed regression model is

$$
\eta_{ij} = \log(\mu_{ij}) = x_{ij}^{T}\beta + u_j \tag{7}
$$

$$
\xi_{ij} = \log\left(\frac{\pi_{ij}}{1 - \pi_{ij}}\right) = z_{ij}^{T}\gamma + w_j \tag{8}
$$

Where $\beta$ and $\gamma$ are the corresponding $(p+1)\times 1$ and $(q+1)\times 1$ vector of regression coefficients. The random effects $u_j$ and $w_j$ are assumed to be independent and normally distributed with a mean of 0 and variance of $\sigma_u^2$ and $\sigma_w^2$, respectively [19].

## Result

Out of the 14,370 mothers in the country, 7720 (54%) of them never faced any child death, while the remaining 6650 have at least one child death. This indicates zero outcomes were large in number. The histograms are highly picked at the beginning (the zero values). However large observations (i.e. large numbers of under-five deaths per mother) are observed less frequently. This leads to a positively skewed distribution. Additional screening of the number of child deaths showed that the variance (1.697) was greater than the mean (0.9) indicating over-dispersion. This indicates that the data could be fitted better by negative binomial hurdle which takes into account excess zeroes (Fig 1).

### Child mortality and its socio-demographic and economic features

Below are summarized impacts of socioeconomic, demographic, health, and environmental-related factors on child death per mother. The majority, 54.1% of deaths occur with uneducated mothers, while for mothers with secondary education and above, the deaths account for 19.7%. The majority (53.4%) occurred among uneducated fathers and 25.3% of the deaths occurred with fathers who had attained secondary and above. Children born in rural areas recorded the highest percentage of deaths, while the death from the urban area was low. The highest percent of child deaths was observed among children having birth orders of four and above (54.0%). The lowest percent of child death was observed in mothers in their first birth order (37.8%). The percentages of death for males and females children were 47.9% and 44.5% respectively. The majority (50%) of child deaths were attributed to poor women. Besides, the highest (58.5%) percent of child deaths occurred among mothers who did not receive any antenatal check during pregnancy. In another respect, working fathers have a lower percent of child deaths (45.4%) as compared to non-working fathers (50.4%). Furthermore, the chi-square test of association revealed that the mother's occupation, current marital status, source of drinking water, type of toilet facility, whether they are currently breastfeeding and place of

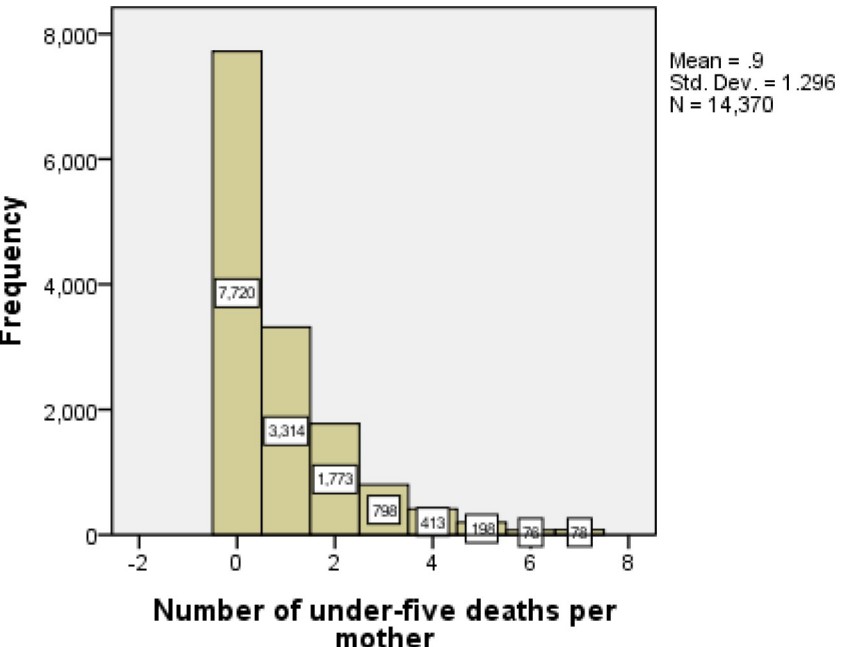

**Fig 1. The number of under-five deaths per mother.**

residence did not was not significantly associated with child death while other variable did (Table 1).

## Model selection criteria

As compared to other models, the negative binomial hurdle regression model has a smaller value of deviance AIC and BIC than the other model. Consequently, we selected the Negative binomial hurdle regression model as the best model for fitting child mortality in Ethiopia (Table 2).

## Factors associated with child mortality in Ethiopia

Table 3 presents summaries from the negative binomial hurdle model. The result of this model gave the fixed and random effects for both the negative binomial and logit components. The negative binomial component shows the Incidence of Relative Riske (IRR) or the severity of child mortality. Child vaccination has a significant impact on the incidence rate of non-zero child death per mother. More particularly, the rate of incidence of non-zero child death for vaccinated children decreased by 26.5 percent (IRR = 0.735, 95%CI: 0.647, 0.834) as compared with non-vaccinated children. For every unit increased in family size, the rate of non-zero under-five death per mother was decreased by 3.2 percent (IRR = 0.968, 95%CI: 0.956, 0.980). Similarly, for a yearly increase in the age of the mother the rate of non-zero child death increased by 5.2 percent (IRR = 1.052, 95%CI: 1.047, 1.056).

The incidence rate of non-zero child death among mothers who had antenatal checks four times and above during the pregnancy was 0.841(IRR = 0.841, 95%CI: 0.737,0.960) times lower compared with mothers who have not received any antenatal check. The rate of non-zero child death among children born 37 months and above after the previous birth decreased by 27 percent (IRR = 0.728, 95%CI: 0.676, 0.783) as compared with children born less than 24 months after the previous birth. The rate of non-zero child death for mothers with primary

**Table 1. Summary statistics of child mortality for selected variable included in the analysis.**

| Variable | Category | Child death | | Total | $X^2$ value (p-value) |
|---|---|---|---|---|---|
| | | Yes | Percent | | |
| Fathers education level | No education | 4,403 | 53.4 | 8,250 | |
| | Primary | 1,736 | 42.3 | 4,101 | 149.64(<0.001) |
| | Secondary and above | 511 | 25.3 | 2,019 | |
| Child Twin | Single | 6,225 | 45.1 | 13,813 | 210.11(<0.001) |
| | Multiple | 425 | 76.3 | 557 | |
| Place of delivery | Home | 5,999 | 55.1 | 10,884 | |
| | Public sector | 578 | 18.6 | 3,107 | 410.51(<0.001) |
| | Private sector | 73 | 19.3 | 379 | |
| Mother's occupation | Housewife | 3,552 | 43.5 | 8,169 | 2.80(0.91) |
| | Had working | 3,098 | 50.0 | 6,201 | |
| Husband/partner's occupation | Not Working | 1,300 | 50.4 | 2,579 | 156.56(<0.001) |
| | Had Working | 5,350 | 45.4 | 11,791 | |
| The educational level of the mother | No Education | 5,378 | 54.1 | 9,932 | |
| | Primary | 1,027 | 32.1 | 3,197 | 32.91(<0.001) |
| | Secondary and above | 245 | 19.7 | 1,241 | |
| Current marital status | Others | 692 | 53.9 | 1,284 | |
| | Married | 5,958 | 45.5 | 13,086 | 0.61 (0.431) |
| Source of drinking water | Piped | 563 | 29.5 | 1,906 | |
| | Others | 6,087 | 48.8 | 12,464 | 0.71 (0.400) |
| Type of toilet facility | Use toilet | 3,506 | 44.3 | 7,911 | |
| | No any kind toilet | 3,144 | 48.7 | 6,459 | 0.01 (0.938) |
| Currently breastfeeding | No | 3,858 | 59.9 | 6,436 | 2.20(0.140) |
| | Yes | 2,792 | 35.2 | 7,934 | |
| Birth order number | First | 1,213 | 37.8 | 3,206 | |
| | 2–3 | 1,951 | 41.5 | 4,705 | 232.943(<0.001) |
| | 4 and above | 3,486 | 54.0 | 6,459 | |
| Place of residence | Urban | 816 | 32.5 | 2,512 | 0.61(0.43) |
| | Rural | 5,834 | 49.2 | 11,858 | |
| Religion | Orthodox | 1,839 | 43.7 | 4,208 | |
| | Muslim | 3,564 | 48.5 | 7,348 | 30.22(<0.001) |
| | Others | 1,247 | 44.3 | 2,814 | |
| Contraceptive use | No | 5,471 | 49.8 | 10,976 | 237.98(<0.001) |
| | Yes | 1,179 | 34.7 | 3,394 | |
| Vaccination of child | No | 6,053 | 50.9 | 11,881 | 201.69(<0.001) |
| | Yes | 597 | 24.0 | 2,489 | |
| Wealth index | Poor | 3,896 | 49.8 | 7,820 | |
| | Medium | 970 | 47.6 | 2,036 | 123.89(<0.001) |
| | Rich | 1,784 | 39.5 | 4,514 | |
| Sex of child | Male | 3,642 | 47.9 | 7,607 | 1.67 (0.195) |
| | Female | 3,008 | 44.5 | 6,763 | |
| Mother's age group | Below 20 years | 145 | 14.9 | 974 | |
| | 20–29 years | 1,377 | 26.5 | 5,201 | 185.28(<0.001) |
| | 30–39 years | 2,935 | 52.6 | 5,582 | |
| | 40+ year | 2,193 | 83.9 | 2,613 | |

*(Continued)*

**Table 1.** (Continued)

| Variable | Category | Child death | | Total | X² value (p-value) |
|---|---|---|---|---|---|
| | | Yes | Percent | | |
| Number of antenatal Visits | No Visits | 5,654 | 58.5 | 9,658 | |
| | 1–3 | 511 | 24.4 | 2,092 | 198.58(<0.001) |
| | 4 and above visited | 485 | 18.5% | 2,620 | |
| Preceding birth interval | 0–24 months | 3,669 | 51.5 | 7,129 | |
| | 25–36 months | 1,642 | 48.2 | 3,407 | 181.00(<0.001) |
| | >36 months | 1,339 | 34.9 | 3,834 | |

education was 0.785(IRR = 0.785, 95%CI: 0.713, 0.864) times lower compared to women with no formal education. Likewise, the incidence rate of non-zero child death for fathers with secondary education and above was 0.695(IRR = 0.695, 95%CI: 0.594, 0.814) times lower compared to fathers with no formal education.

The rate of non-zero child deaths with children's birth order 4 and above increased by 48.7 percent (IRR = 1.487, 95%CI: 1.373, 1.612) compared with the first birth order. The incidence rate of non-zero child death for mothers who used contraceptives was about 0.885 (IRR = 0.885, 95%CI: 0.814, 0.962) times lower than mothers who did not use a contraceptive. The rate of non-zero child death for children born in the private health facility was 0.609 (IRR = 0.609, 95%CI: 0.405, 0.916) times lower than that of children born at home. The incidence rate of non-zero child death in multiple births was 1.355 (IRR = 1.355, 95%CI: 1.249, 1.471) times greater compared with that of a single birth. The rate of non-zero child deaths for mothers older than16 years decreased by 29 percent (IRR = 0.711, 95%CI: 0.674, 0.750) compared with mothers older than 17 years (Table 3).

The Bernoulli or logit part used to show the likelihood of child mortality on the household level. We observed that the estimated odds of the number of child death becomes zero with vaccinated children were 1.825 (AOR = 1.825, 95%CI: 1.614, 2.064) times the non-vaccinated children. An increase in family size by 1 result in the estimated odds that the number of child death becomes zero was increased by 25.5 percent (AOR = 1.255, 95%CI: 1.224, 1.286). Similarly as the age of mother increase by a year the estimated odds that the number of child death becomes zero decreases by 18 percent (AOR = 0.822, 95%CI: 0.815, 0.829). The estimated odds that the number of child death becomes zero with mothers who made antenatal care of 4 visits and above was 2.390(AOR = 2.390, 95%CI: 2.066,2.764) times that of mothers who did not any antenatal visit.

The estimated odds that the number of child death becomes zero for children born with a preceding birth interval of 37 months and above was 3.523(AOR = 3.523, 95%CI: 3.134, 3.960) times that of children born with a preceding birth interval of fewer than 24 months. The odds of the number of child death becomes zero with children born from fathers who work is 0.721 (AOR = 0.721, 95CI%: 0.633, 0.821) times that of fathers without work. The estimated odds the

**Table 2. Model selection criteria for the multilevel count regression models.**

| Model | Deviance | AIC | BIC |
|---|---|---|---|
| ZIP | 29534 | 29578 | 29744 |
| ZINB | 29368 | 29414 | 29588 |
| PH | 28222 | 28306 | 28624 |
| NBH | 28183 | 28269 | 28595 |

Table 3. Fixed and random effects estimates with corresponding 95 confidence intervals (CI) from the negative binomial hurdle model of child mortality.

| Parameters | Negative binomial | Bernoulli |
|---|---|---|
| | IRR (95% CI for IRR) | OR(95 CI% for OR) |
| Fixed effects | | |
| **Vaccination child** | | |
| No | 1 | 1 |
| Yes | 0.735(0.647,0.834)* | 1.825 (1.614,2.064)* |
| **Family size** | 0.968(0.956,0.980)* | 1.255 (1.224,1.286)* |
| **Age of mother** | 1.052(1.047,1.056)* | 0.822 (0.815,0.829)* |
| **Antenatal care visit** | | |
| No | 1 | 1 |
| 1–3 | 0.841(0.737,0.960)* | 2.013(1.757,2.306)* |
| 4 and plus | 0.814(0.702,0.944)* | 2.390(2.066,2.764)* |
| **Previous birth interval** | | |
| ≤24 months | 1 | 1 |
| 25–36 months | 0.836(0.787,0.889)* | 1.766(1.579,1.974)* |
| 37 and plus | 0.728(0.676,0.783)* | 3.523(3.134,3.960)* |
| **Contraceptive use** | | |
| No | 1 | 1 |
| Yes | 0.885(0.814,0.962)* | 1.354(1.202,1.525)* |
| **Father's education** | | |
| No education | 1 | 1 |
| Primary | 0.945(0.881,1.014) | 0.970(0.867,1.085) |
| Secondary and plus | 0.695(0.594,0.814)* | 1.211(1.014,1.446)* |
| **Mother's education** | | |
| No education | 1 | 1 |
| Primary | 0.785(0.713,0.864)* | 1.145(1.011,1.298)* |
| Secondary and plus | 0.787(0.614,1.009) | 1.433(1.136,1.806)* |
| **Father occupation** | | |
| No | 1 | 1 |
| Had Working | 1.125(1.049,1.206)* | 0.721(0.633,0.821)* |
| **Place of delivery** | | |
| Home | 1 | 1 |
| Public sector | 0.927(0.809,1.061) | 2.053(1.785,2.361)* |
| private sector | 0.609(0.405,0.916)* | 1.947(1.419,2.673)* |
| **Child Twin** | | |
| Single | 1 | 1 |
| Multiple | 1.355(1.249,1.471)* | 0.256(0.197,0.333)* |
| **Age of mother at first birth** | | |
| ≤ 16 year | 1 | 1 |
| 17 and plus year | 0.711(0.674,0.750)* | 3.004(2.722,3.314)* |
| **Birth order** | | |
| First | 1 | |
| 1–3 | 1.372(1.262,1.491)* | |
| 4 and above | 1.487(1.373, 1.612)* | |
| **Religion** | | |
| Orthodox | 1 | |
| Muslim | 1.255(1.129,1.394)* | |
| Others | 1.104(0.978,1.246) | |

(*Continued*)

**Table 3.** (Continued)

| Parameters | Negative binomial | Bernoulli |
|---|---|---|
| | IRR (95% CI for IRR) | OR(95 CI% for OR) |
| **Random effect** | | |
| **Between-Enumeration Area variance ($\hat{\sigma}^2_{u0}$)** | 0.526(0.474,0.584)* | 0.691(0.624, 0.766)* |

1: reference category of the categorical variable.

* Significant at 0.05 level of significance.

number of child death with mothers who use contraceptives is 1.354(AOR = 1.354, 95%CI: 11.202, 1.525) times mothers who did not use a contraceptive. The estimated odds that the number of child death becomes zero with children who are born in the public health sector was 2.053(AOR = 2.053, 95%CI: 1.785, 2.361) times that of children born at home. The estimated odds the number of child death becomes zero with mothers who attend primary education was 1.145 (AOR = 1.145, 95%CI: 1.011, 1.298) times mothers who did not attend any formal education.

The estimated odds the number of child death becomes zero among children born in multiple births decreased by 74.4 percent (AOR = 0.256, 95%CI: 0.197, 0.333) as compared to children born in a single birth. As could be seen in Table 3, the variance components for the random effects showed that significant variation of child mortality between enumeration area is estimated to be 0.526 (95% CI: 0.474, 0.584) while the severity of the variance was 0.691 (95% CI: 0.624, 0.766) (Table 3).

## Spatial distribution of the under-five mortality rate

A total of 645 enumeration areas (clusters) were included in the analysis. Spatial distribution of under-five mortality counts and the residuals were mapped (Fig 2). The crude mortality and residuals were computed based on each enumeration area (EA), which were then merged with the shapefiles and mapped to present the regional disparities. Hence, the regional (EA) counts

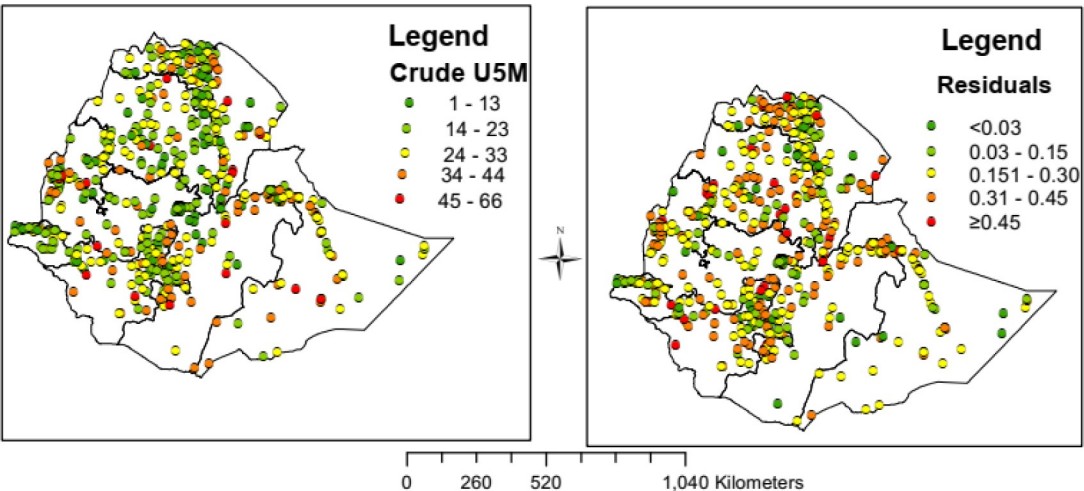

**Fig 2. Spatial distribution of U5 crude mortality and the residual in Ethiopia, EDHS 2016 perspective: Each dot on the map represents one enumeration area.**

were extracted and plotted in the form of geometric lines to show the burden of under-five mortality for the study period. Both the maps showed that there were notable inequalities in crude mortality among the regions/clusters of the country.

The two maps indicated that there is wide variability in U5 mortality rate among the 645 enumeration areas (Fig 2)

## Discussion

The child mortality rate in Ethiopia declined over time. The death rate in the period2000 2005 20011 and 2016 was 166 123 88 and 67 deaths per 1000 live births respectively. This multilevel analysis in children's mortality rate showed improving (decreasing) trend from 166 children per 100 live births in 2000 to about 67deaths per 1000 live births in2016. It reduced at the rate of 99deaths per 1000 live births for a period of 16 years. This study set out to develop a predictive model and investigate modifiable risk factors of mortality for under-five in Ethiopia. Moreover, it examined Enumeration Area variations in under-five mortality that was not studied so far and that could not be explained by the available risk factors. It examined the influence of particular social-economic and demographic characteristics of mothers on child mortality in Ethiopia. Results showed several factors contributed to child mortality. It was observed that out of 14730 women in the data 7720 (53.7) of them did not experience child mortality. Level of parental education emerged as a strong predictor of child mortality that decreases with an increase in parental education indicating that improved parental education minimizes the number of child mortality. This supports previous findings where increased maternal and father education lowered child mortality [13, 27–31]. Thus, improving the educational background of parents advantages to themselves their children and the community as a whole [32, 33]. Educated parents are more likely to be aware of health care utilization to themselves their children and the community [30, 32, 34]. Child death with multiple births is higher relative to singleton ones. Multiple births have a lower weight due to nutritional intake competition [10, 12, 13, 29]. The result also showed that child mortality decreased with an increase in the length of the preceding birth interval, which confirms with findings from previous studies in Ethiopia that employed survival analysis and binary logistic regression models and where the preceding birth interval is negatively associated with mortality of a child [12–14].

The finding revealed that the death of children from mothers who use contraceptives is significantly less compared to the death of children from mothers who did not use a contraceptive. The size of a family is positively associated with a mortality rate of a child. These variables are in one way or another associated with family planning. This finding is consistent with from previous studies that investigated under-five mortality in developing countries especially in Sub Saharan African where are born from mothers that don't use contraceptive and that have a large family size which in turn associated with an increase in the odds of under-five mortality [14, 35]. Vaccinated children have are a lower risk of mortality compared to non-vaccinated children and this finding is consistent with those of previous studies [10, 30, 36, 37].

Mother's age at first birth is negatively correlated with child mortality that decreased the risk of child mortality as an increase in mother's age at first birth. The estimated result also shows that increases mothers' age at first birth reduced the risk of child mortality and mothers who gave birth to their first child at a younger age face higher child mortality risk which is similar to the previous studies conducted by different scholars in developing countries including Ethiopia, Nigeria and Bangladesh [12, 14, 28, 29, 35]. Studies also reported that similar to the findings [28, 29] for every unit increase in the age of the mother, child mortality increases.

This study indicated that mothers/children raised in urban areas were less likely to die due to a lack of better health care access and other important services crucial for the health of the child [36, 38].

The study showed that children from working fathers have a higher risk of mortality than those from non-working fathers. This finding is consistent with where the increase in the number of antenatal visits during pregnancy reduces child mortality, which is also supported by the previous research [14]. Children born in the public and private sectors are at lower risk than those born at home. This might be due to better health care and attention received during and after delivery. This has been corroborated by different studies [12, 14, 39]. The study also revealed that the size of a household is important in affecting the number of child mortality. The mortality decreased significantly as household size increased, a finding which is unexpected and inconsistent with finding from previous studies [37]. Consequently, further research is needed to determine the relationship between the variables. Birth order increased child mortality, and this result is consistent with the literature reviewed and contribution from different studies on birth order [10, 12, 40]. The prevalence and severity implied that there existed heterogeneity of child death in terms of their enumeration area through each child shared the same covariate value. Distribution of socioeconomic resources which largely affect the health condition of the population showed (enumeration area disparity in health service allocation and nutrition). This is consistent with previous findings in which that geographical location influenced health outcomes [28, 31, 34, 36, 37, 41].

## Strengths and weaknesses

The strength of the DHS is its representativeness, its use of starts it's multistage probabilistic sampling for selecting clusters and households from different geographical territories so that quality data are collected about children at household and community levels. This cross-sectional study involves a randomly selected large sample so that the findings could be generalized to the studied population. To increase reliability and avoiding missing data, data collectors and interviewers received pertinent training. Quantitative survey data are often under-count (hard to reach the whole EAs/groups) and owing to its cross-sectional nature, it is difficult to measure the causal effects, and it is not possible to know whether the data are time-dependent or not. The other limitation of the study (dataset) is that it includes limited measurement of adult health outcomes of women and men population and does not cover even communicable and non-communicable diseases [42].

## Conclusions

Mortality of the under-five in developing nations like Ethiopia is still a public health problem, and this study tried to identify the key determinants of and assessing the enumeration area variation of child death in the country. A multilevel NBH modeling approach allowed the determination of unobserved enumeration area differences in under-five children's mortality rates that cannot be addressed through a single-level approach. The descriptive results showed that 53.7% of mothers did not experience the death of under-five while 46.3% of them lost at least one child. The findings indicated an enumeration area variation in child mortality. The high-risk factors associated with under-five mortality were higher order of birth of the child, multiple twins, mothers who have their first birth at age 16 or below, not using the contraceptive, no getting the child vaccinated, rural women, uneducated mother, smaller family size, older age, uneducated father's and mother 's antenatal visit of healthcare. Therefore Policies and programs aimed at addressing enumeration area variations in child mortality must be formulated and their implementation must be vigorously pursued. The government should give more

attention to regions with a high child mortality rate. Furthermore, programs of educating mothers on the benefits of the antenatal check-up birth spacing or birth interval having vaccination for the child and safer place of child delivery need to be considered to reduce child death.

## Acknowledgments

The authors would like to thank the Central Statistical Agency of Ethiopia for making the data freely available for research purposes. The manuscript was language edited by Berhanu Engidaw (Assistant professor), English department, Bahir Dar University.

## Author Contributions

**Conceptualization:** Setegn Muche Fenta, Haile Mekonnen Fenta.

**Data curation:** Setegn Muche Fenta.

**Formal analysis:** Setegn Muche Fenta, Haile Mekonnen Fenta.

**Investigation:** Setegn Muche Fenta.

**Methodology:** Setegn Muche Fenta, Haile Mekonnen Fenta.

**Resources:** Setegn Muche Fenta.

**Software:** Setegn Muche Fenta, Haile Mekonnen Fenta.

**Validation:** Setegn Muche Fenta.

**Writing – original draft:** Setegn Muche Fenta.

**Writing – review & editing:** Haile Mekonnen Fenta.

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
