## [Decision Letter · Decision Letter 0]

2 Apr 2020

PONE-D-19-33030

Risk Factors of Child Mortality in Ethiopia: Application of Multilevel Two-Part Model

PLOS ONE

Dear Mr. Fenta,

Thank you for submitting your manuscript to PLOS ONE. After careful consideration, we feel that it has merit but does not fully meet PLOS ONE’s publication criteria as it currently stands. Therefore, we invite you to submit a revised version of the manuscript that addresses the points raised during the review process.

The manuscript has been evaluated by three reviewers; their comments are available below.

The reviewers have raised a number of major concerns that need attention in a revision. The reviewers note that the rationale for the study needs to be more clearly articulated and a stronger case made for the evidence gap that is being addressed by this work. The reviewers request clarifications about the population included and the variables studied, and note that improvements are needed in the Discussion section to ensure the limitations of the work are discussed and that the findings are adequately interpreted and the contributions of the work to the field clearly outlined.

The reviewers also note that improvements are needed to the written language, we note that the English requires substantial edits. Please ensure that you copyedit the manuscript prior to resubmission, please note that further consideration is dependent on the submission of a revised manuscript where adequate improvements have been made to the written language.

Could you please revise the manuscript to address the items raised?

We would appreciate receiving your revised manuscript by May 16 2020 11:59PM. Please include the following items when submitting your revised manuscript:

We look forward to receiving your revised manuscript.

Kind regards,

Iratxe Puebla

Deputy Editor-in-Chief, PLOS ONE

Journal Requirements:

2. Please ensure you have thoroughly discussed any potential limitations of this study within the Discussion section.

Reviewers' comments:

Reviewer's Responses to Questions

**Comments to the Author**

1. Is the manuscript technically sound, and do the data support the conclusions?

Reviewer #1: Yes

Reviewer #2: Partly

Reviewer #3: Partly

2. Has the statistical analysis been performed appropriately and rigorously? 

Reviewer #1: Yes

Reviewer #2: Yes

Reviewer #3: No

3. Have the authors made all data underlying the findings in their manuscript fully available?

Reviewer #1: Yes

Reviewer #2: No

Reviewer #3: Yes

4. Is the manuscript presented in an intelligible fashion and written in standard English?

Reviewer #1: Yes

Reviewer #2: No

Reviewer #3: No

5. Review Comments to the Author

Reviewer #1: The manuscript describes an interesting study on risk factors of child mortality in Ethiopia based on multilevel two-part models using secondary data from the 2016 Ethiopian Demographic and Health Survey (i.e. the most current DHS data in the country). Studies of this nature are critical for informed public health policy/intervention strategies. However, the authors failed to demonstrate that there is a risk factor gap for child mortality in Ethiopia to warrant this study. They were rather making a case for statistical methods components (multilevel two-part model).

Also, the factors identified in this study as predictors of child mortality have been established in similar populations in Africa using both single and multilevel methods, including spatial mapping. The authors should therefore make a case for the complex model as oppose to the relatively simpler model that achieved the same purpose: risk factor identification.

See more details below:

Introduction:

a) The introduction to the study appears good. However, this section will benefit from proofread. For example, statement like “Child mortality wants serious attention from …” should be rephrased.

b) Also, authors should briefly explain what has been done so far by government and other stakeholders to address the problem of child mortality in Ethiopia, and why the problem persists to the extent that Ethiopia is among the top 6 countries with high child mortality rates.

Methods:

c) The methods employed by the authors in the analysis of the data appears sound.

d) The authors should indicate the number of regions that were used in their study since they are interested in ‘between-region variance’. This is critical for the multilevel part because the number of groups (regions) influence the precision of the model parameters. Thus, small number of groups (regions) can substantially affect the accuracy and the interpretability of results from multilevel model part like the one used in this study. For example, Maas and Hox (2005) have shown that small number of groups at the higher level (i.e. sample of 50 groups or less) will result in biased estimates for the higher-level (i.e. region in this case) standard errors.

e) How did the authors adjust for the sampling weight in their study? The Ethiopian Demographic and Health Survey (DHS), just like any other DHS include an inherent sampling weight so the authors should discuss how they account for this in their analysis. Not accounting for this could bias the model parameters and its resultant misleading inferences.

Reference

Maas, CJM & Hox JJ (2005). Sufficient sample sizes for multilevel modelling. Methodology, 1, 86-92.

f) The factors identified in this study as predictors of child mortality have been established in similar populations in Africa using both single and multilevel methods, including spatial mapping. The authors should therefore make a practical rather than a theoretical case for the complex model (Hurdle Negative Binomial) as oppose to the relatively simpler models (single, multilevel, and spatial models) that previously achieved the same purpose: risk factor identification.

Results:

g) The results presented are satisfactory.

h) Comments under the methods section could be considered to improve the message in this section.

Discussion and conclusion:

i) The discussion and the conclusion presented are supported by the data.

j) There is the need to proofread the entire manuscript for clarity and understanding.

k) Using DHS data comes with some limitations, but the authors failed to state what the limitations and strengths of their study are. This must be provided.

Reviewer #2: Comments

Topic: Risk Factors of Child Mortality in Ethiopia: Application of Multilevel Two-Part Model

General comments

This a good paper whose subject is quite relevant to researchers, programmers and policy makers involved in understanding and increasing quality of neonatal health care. There are however a few comments on the manuscript which if corrected should make the paper acceptable for publication.

Major Compulsory Revisions

• The study examined the influence of particular social, economic and demographic characteristics of mothers on child mortality in Ethiopia. And also the findings indicate there was a variation of child mortality from region to region. This can be important in public health area as part of the evaluation of planned interventions, as well as for policymakers for indicating future directions. But this is already known in Ethiopia, and a few of references support this. General findings are well known also for the Ethiopian community. What the findings add to what already known?

• Information concerning potential variables related to child mortality and then analyzed and discussed. No specific suggestion was reported for each single Region taking into account difference in child mortality.

• Discussions: It is quite poor and repeats a lot of known facts without making any point as to how this current study contributes to the discussion. A lot of results are repeated in the discussion. What are the innovative ideas, for scale up and ensure quality and safe services? Formulate clear what is innovating idea in the study.

• Conclusion section: there is significant disconnect between the results presented and the conclusions made. There was no evidence in the results or anywhere else that they looked at the possible barriers and strategies for that country under question. They can suggest but not make a hard conclusion that those strategies would work or hinder.

Reviewer #3: PLOS ONE

Risk Factors of Child Mortality in Ethiopia: Application of Multilevel Two-Part Model

Abstract

What do you mean by “being contraceptive use”: At which stage? Before or after the child’s death?

What do you mean by “primary level of educated mother”?

What is “being had working father”

There is no result in the abstract to justify “The findings indicate a significant regional variation of child mortality in Ethiopia.”

Objective is clear

Methods

Page 4

“The women were interviewed by distributing questioners”?

What are the justifications for including the Independent Variables?

Page 5

Citations such as “The ZIP model, introduced by [17],…..” is wrong. Replace here and elsewhere as “Lambert et al. introduced the ZIP model [17],….” or “The ZIP model, introduced by Lambert et al. [17],…..”

mothers who may not be dead her child).?

Page 7

The Poisson Hurdle and Negative binomial Poisson mixed regression model is (7) and (8) respectively?

Check : distributed with mean 0 and variance 2u ��and 2 w ��,

And 7720 of them lost 7 at least one child.?

Page 8

The subsection “The trend of child mortality in Ethiopia (2000-2016)” and Figure 2 are not results of the current study. These should be incorporated into the introduction or discussion section.

Table 1

Delete all ‘%’ in the Table except in the headings

The column “No(%)” is redundant. Delete

What is p-value of <,0.001? delete the comma sign

The study population is not very clear.

I presume this study is about all deaths among all children to a woman who had ever given birth. Or is it all deaths among children born to a woman within the 5-years preceding the survey. Which data did the author used? The women data? Birth recode data? Or children recode data?

ALL these must be clearly stated under methods and Data

More importantly, since all deaths at a particular time is been studied, which birth or child was used to classify the variables in Table 1. For instance, the variable Child Twin has single and multiple. Assuming a mother had a twins birth and had a single birth thereafter, which of the births was used to classify the twin status. Same thing applies to place of delivery, Birth order number, Currently breastfeeding, Contraceptive use, Vaccination of child, Sex of child, Number of antenatal Visits and Preceding birth interval.

Variables here should be strictly mothers characteristics. The only child characteristic of interest here sis the outcome variable which is “child status: Alive or death”

This brings another issue, at which age of the children did the authors classified a child as dead or alive, since there are multiple children per woman in some cases

Table 2

The results here are not reliable because some of the variables in the models are inappropriate for the reasons stated earlier. Addition to that list is “family size”. This is not static but changes as there are more births or deaths. How can such predict death?

I stopped the review here

General comments……..

Good study design but the authors did not pay attention to basic details

Too numerous typos, incomplete statements etc.

They had reported that 7720 women had lost at least one child. And also claimed that 7720 women had lost 7 children!!!

6. PLOS authors have the option to publish the peer review history of their article (what does this mean?). If published, this will include your full peer review and any attached files.

Reviewer #1: Yes: Justice Moses K. Aheto

Reviewer #2: No

Reviewer #3: No

---

## [Author Response · Author response to Decision Letter 0]

2 May 2020

Reviewer 3

Thank you so much for your question. Generally, it is defined as “Women use contraceptive before childbirth as well as after birth”

Mothers who have attained only primary school (1 to 8 grade level in the country context)

We have corrected all the comments based on your suggestion. Some points were typing error and it is corrected right now. 

Thank you once again, it is incorporated in the discussion part 

Yes, removed and the p-value<0.001, in the software result it was highly significant (0.000) and in the scientific format, we make it <0.001, and the comma is removed. 

Yes, we considered all children ever born from (birth record data from DHS) and we count the number of U5 death per each women

This is a very important and interesting question. Since children from the same mother share the same characteristics so that we addressed this situation by incorporating the complex sampling method (weighting approach) and considered all the data. 

Here our interest is counting the number of under-five child mortality /death per woman, but individual/child variables have also its impact on child mortality.

we got your comments are very valuable. we answer the specific question here. The manuscript is well edited by professionals

---

## [Decision Letter · Decision Letter 1]

20 May 2020

PONE-D-19-33030R1

Risk factors of child mortality in Ethiopia: Application of multilevel two-part model

PLOS ONE

Dear Mr Fenta,

Thank you for submitting your manuscript to PLOS ONE. After careful consideration, we feel that it has merit but does not fully meet PLOS ONE’s publication criteria as it currently stands. Therefore, we invite you to submit a revised version of the manuscript that addresses the points raised during the review process.

Thank you for responding to some of the comments raised in the first round of review. After perusing your revised manuscript, the Editors and the Reviewers were of the considered opinion that not all concerns initially raised were satisfactorily addressed and some responses were put together to cover multiple queries, especially for Reviewer #3. It is required that you provide point-by-point responses to each query raised in the initial revision, including the new queries. 

Rejoinder

Following up on your initial responses, the Editors have a rejoinder concerning the use of only 11 regions (i.e. the total available in Ethiopia) as the higher-level (grouping variable) for your multilevel model presented in Table 3. Please, address the query below in your revision (also, see Additional Editor Comments section):

You indicated that there is only a total of 11 regions in Ethiopia and that you used all of them as the grouping variable (high-level) in the multilevel model. The small number of regions used in this study could potentially bias the results presented in Table 3. You are requested to conduct a sensitivity analysis for the negative binomial hurdle model results presented in Table 3 to determine whether or not the small group size could affect the estimates. In case you are unable to do this, an alternative will be for you to provide both 95% and 99% confidence intervals for the negative binomial hurdle model results presented in Table 3 and discuss both.

You should also provide regional crude U5 mortality map for Ethiopia based on the data, and also provide the regional residual (regional random effects) map based on the negative binomial hurdle model results presented in Table 3 for better understanding of the regional (spatial) distribution of crude U5 mortality rates and the residual regional effects in Ethiopia.

We would appreciate receiving your revised manuscript by 20 June 2020. To enhance the reproducibility of your results, we recommend that if applicable you deposit your laboratory protocols in protocols.io, where a protocol can be assigned its own identifier (DOI) such that it can be cited independently in the future. For instructions see: http://journals.plos.org/plosone/s/submission-guidelines#loc-laboratory-protocols

We look forward to receiving your revised manuscript.

Kind regards,

Justice Moses K. Aheto, HND, BSc, MSc, PhD

Academic Editor

PLOS ONE

Additional Editor Comments (if provided):

Following up on your initial response, the Editors have a rejoinder concerning the use of only 11 regions (i.e. the total available in Ethiopia) as the higher-level (grouping variable) for your multilevel model. Please, address the queries below in your revision:

You indicated that there is only a total of 11 regions in Ethiopia and that you used all of them as the grouping variable (high-level) in the multilevel model. The small number of regions used in this study could potentially bias the results presented in Table 3. You are requested to conduct a sensitivity analysis for the negative binomial hurdle model results presented in Table 3 to determine whether or not the small group size could affect the estimates. In case you are unable to do this, an alternative will be for you to provide both 95% and 99% confidence intervals for the negative binomial hurdle model results presented in Table 3 and discuss both.

You should also provide regional crude U5 mortality map for Ethiopia based on the data, and also provide the regional residual (regional random effects) map based on the negative binomial hurdle model results presented in Table 3 for better understanding of the regional (spatial) distribution of crude U5 mortality rates and the residual regional effects in Ethiopia.

Reviewers' comments:

Reviewer's Responses to Questions

**Comments to the Author**

1. If the authors have adequately addressed your comments raised in a previous round of review and you feel that this manuscript is now acceptable for publication, you may indicate that here to bypass the “Comments to the Author” section, enter your conflict of interest statement in the “Confidential to Editor” section, and submit your "Accept" recommendation.

Reviewer #3: (No Response)

2. Is the manuscript technically sound, and do the data support the conclusions?

Reviewer #3: No

3. Has the statistical analysis been performed appropriately and rigorously? 

Reviewer #3: Yes

4. Have the authors made all data underlying the findings in their manuscript fully available?

Reviewer #3: Yes

5. Is the manuscript presented in an intelligible fashion and written in standard English?

Reviewer #3: No

6. Review Comments to the Author

Reviewer #3: The submission did not contain responses to my comments in revision 1. These must be stated one by one.

The authors refused to adopt most of the suggested changes without rebuttal.

Page 9

Why did you highlight “Eighty-eight (87.7%)”

Table 1: The column “No(%)” must be removed. It adds no information. Rather…Total number for each category is more informative. Replace accordingly

Change “X2 test (p-value” to ” X2 value (p-value)”

Table 3…all your confidence intervals are joined together, infact there are no CI….

Change “(95 CI for IRR)” to “(95% CI for IRR)”

7. PLOS authors have the option to publish the peer review history of their article (what does this mean?). If published, this will include your full peer review and any attached files.

Reviewer #3: No

---

## [Author Response · Author response to Decision Letter 1]

2 Jun 2020

Point 1: You indicated that there is only a total of 11 regions in Ethiopia and that you used all of them as the grouping variable (high-level) in the multilevel model. The small number of regions used in this study could potentially bias the results presented in Table 3. You are requested to conduct a sensitivity analysis for the negative binomial hurdle model results presented in Table 3 to determine whether or not the small group size could affect the estimates. In case you are unable to do this, an alternative will be for you to provide both 95% and 99% confidence intervals for the negative binomial hurdle model results presented in Table 3 and discuss both.

Answer: 

This is a very important question. We have read literatures and textbooks related to the multilevel modeling. We have got a very interesting point that “ the minimum number of the higher level is at least 50” . In this study we have only 11 regions. However, the EDHS dataset was collected from 645 enumeration areas in the country. We authors reconsider the multilevel modeling by incorporating 645 EAs (instead of 11 regions) as the second level in the analysis.

Point 2: You should also provide regional crude U5 mortality map for Ethiopia based on the data, and also provide the regional residual (regional random effects) map based on the negative binomial hurdle model results presented in Table 3 for better understanding of the regional (spatial) distribution of crude U5 mortality rates and the residual regional effects in Ethiopia.

Answer: This is again very important point to show the hot spot of the incidence. We did and map both the crude U5 moratality rates and the residuals regional effects as figure 2 and figure 3.

Point 3: Why did you highlight “Eighty-eight (87.7%)”

Answer : Thank you. It was typing error, corrected 

Point 4: Table 1: The column “No(%)” must be removed. It adds no information. Rather…Total number for each category is more informative. Replace accordingly

Answer : Very good point, corrected 

Point 6: Change “X2 test (p-value” to ” X2 value (p-value)”

Answer : Thank you, changed 

Point 7: Change “(95 CI for IRR)” to “(95% CI for IRR)”

Answer : Thank you, changed

---

## [Decision Letter · Decision Letter 2]

29 Jun 2020

PONE-D-19-33030R2

Risk factors of child mortality in Ethiopia: Application of multilevel two-part model

PLOS ONE

Dear Dr. Fenta,

Thank you for submitting your manuscript to PLOS ONE. After careful consideration, we feel that it has merit but does not fully meet PLOS ONE’s publication criteria as it currently stands. Therefore, we invite you to submit a revised version of the manuscript that addresses the points raised during the review process.

Thank you for responding to the comments raised in the second round of review. After perusing your revised manuscript, the Editors and the Reviewers were of the considered opinion that your manuscript is sound, but some minor revisions are required for the manuscript to be accepted for publication.

The maps in Figure 2 are not the crude under-five mortality rates and EA random effects as suggested. Note that the models you presented in this study cannot produce any of the maps presented in Figure 2. The maps in Figure 2 are interpolated maps based on some kriging or other spatial prediction approach (making predictions for both sampled and unsampled locations) which required a detailed explanation in the methods section since the centroid of the clusters (EAs - 645 in total) are not spatial polygons but spatial points. The geographic coordinates used in the Demographic and Health Surveys are the centroid of the clusters (EAs) and hence are spatial points.

The map of the crude under-five mortality rates should be the number of under-five deaths (counts) at each centroid location of the cluster (EA) for the 645 EAs used in the study. This is what you were required to produce in line with changing from regions to EAs as the grouping variable. You should also do same for the cluster (EA) random effects based on the negative binomial hurdle model presented in Table 3. Thus, extract the random effects from the negative binomial hurdle model for the 645 EAs and map them at their geographic coordinate locations.

Also, you are required to provide additional details (e.g. access date and URL) for references 2, 7, 8, 9, 19 and 21. Language editing is also required.

We look forward to receiving your revised manuscript.

Kind regards,

Justice Moses K. Aheto, HND, BSc, MSc, PhD

Academic Editor

PLOS ONE

Additional Editor Comments (if provided):

The maps in Figure 2 are not the crude under-five mortality rates and EA random effects as suggested. Note that the models you presented in this study cannot produce any of the maps presented in Figure 2. The maps in Figure 2 are interpolated maps based on some kriging or other spatial prediction approach (making predictions for both sampled and unsampled locations) which required a detailed explanation in the methods section since the centroid of the clusters (EAs - 645 in total) are not spatial polygons but spatial points. The geographic coordinates used in the Demographic and Health Surveys are the centroid of the clusters (EAs) and hence are spatial points.

The map of the crude under-five mortality rates should be the number of under-five deaths (counts) at each centroid location of the cluster (EA) for the 645 EAs used in the study. This is what you were required to produce in line with changing from regions to EAs as the grouping variable. You should also do same for the cluster (EA) random effects based on the negative binomial hurdle model presented in Table 3. Thus, extract the random effects from the negative binomial hurdle model for the 645 EAs and map them at their geographic coordinate locations.

Also, you are required to provide additional details (e.g. access date and URL) for references 2, 7, 8, 9, 19 and 21. Language editing is also required.

Reviewers' comments:

Reviewer's Responses to Questions

**Comments to the Author**

1. If the authors have adequately addressed your comments raised in a previous round of review and you feel that this manuscript is now acceptable for publication, you may indicate that here to bypass the “Comments to the Author” section, enter your conflict of interest statement in the “Confidential to Editor” section, and submit your "Accept" recommendation.

Reviewer #3: All comments have been addressed

2. Is the manuscript technically sound, and do the data support the conclusions?

Reviewer #3: Yes

3. Has the statistical analysis been performed appropriately and rigorously? 

Reviewer #3: Yes

4. Have the authors made all data underlying the findings in their manuscript fully available?

Reviewer #3: Yes

5. Is the manuscript presented in an intelligible fashion and written in standard English?

Reviewer #3: No

6. Review Comments to the Author

Reviewer #3: To review grammatical constructs and provide details of the following references

The following references needs more details eg urls, accessed date

2,7, 8 and 9, 19,21,

7. PLOS authors have the option to publish the peer review history of their article (what does this mean?). If published, this will include your full peer review and any attached files.

Reviewer #3: No

---

## [Editor Report · Decision Letter 3]

27 Jul 2020

PONE-D-19-33030R3

Risk factors of child mortality in Ethiopia: Application of multilevel two-part model

PLOS ONE

Dear Dr. Fenta,

Thank you for submitting your manuscript to PLOS ONE. After careful consideration, we feel that it has merit but does not fully meet PLOS ONE’s publication criteria as it currently stands. Therefore, we invite you to submit a revised version of the manuscript that addresses the points raised during the review process.

Kindly remove Figure 3 from the manuscript and also note that the term is "ordinary Kriging" BUT not "ordinal kriging" as stated at page 15. The removal of Figure 3 is necessary because the ordinary Kriging require detailed explanation (estimation procedure for spatial correlation, range, etc) of this procedure in the methods section, and you must present the map of the prediction variance  and the plot of the known variograms associated with the Figure 3 in the results section for proper evaluation of the interpolated map based on the said ordinary kriging method.  All these are missing in the manuscript presently. Also note that spatial prediction is not the focus of this manuscript hence the earlier request to map only the crude mortality and the residual spatial effect based on your  

We look forward to receiving your revised manuscript.

Kind regards,

Justice Moses K. Aheto, HND, BSc, MSc, PhD

Academic Editor

PLOS ONE

---

## [Editor Report · Decision Letter 4]

31 Jul 2020

Risk factors of child mortality in Ethiopia: Application of multilevel two-part model

PONE-D-19-33030R4

Dear Dr. Fenta,

We’re pleased to inform you that your manuscript has been judged scientifically suitable for publication and will be formally accepted for publication once it meets all outstanding technical requirements.

Kind regards,

Justice Moses K. Aheto, HND, BSc, MSc, PhD

Guest Editor

PLOS ONE
---

## [Editor Report · Acceptance letter]

4 Aug 2020

PONE-D-19-33030R4 

Risk factors of child mortality in Ethiopia: Application of multilevel two-part model 

Dear Dr. Fenta:

I'm pleased to inform you that your manuscript has been deemed suitable for publication in PLOS ONE. Congratulations! Your manuscript is now with our production department. 

Kind regards, 

on behalf of

Dr. Justice Moses K. Aheto 

Guest Editor

PLOS ONE